# The β20–β21 of gp120 is a regulatory switch for HIV-1 Env conformational transitions

Alon Herschhorn[1,2], Christopher Gu[1], Francesca Moraca[3], Xiaochu Ma[4], Mark Farrell[5], Amos B. Smith, III[5], Marie Pancera[6], Peter D. Kwong[6], Arne Schön[7], Ernesto Freire[7], Cameron Abrams [3], Scott C. Blanchard[8], Walther Mothes[4] & Joseph G. Sodroski[1,2,9]

The entry of HIV-1 into target cells is mediated by the viral envelope glycoproteins (Env). Binding to the CD4 receptor triggers a cascade of conformational changes in distant domains that move Env from a functionally "closed" State 1 to more "open" conformations, but the molecular mechanisms underlying allosteric regulation of these transitions are still elusive. Here, we develop chemical probes that block CD4-induced conformational changes in Env and use them to identify a potential control switch for Env structural rearrangements. We identify the gp120 β20–β21 element as a major regulator of Env transitions. Several amino acid changes in the β20–β21 base lead to open Env conformations, recapitulating the structural changes induced by CD4 binding. These HIV-1 mutants require less CD4 to infect cells and are relatively resistant to State 1-preferring broadly neutralizing antibodies. These data provide insights into the molecular mechanism and vulnerability of HIV-1 entry.

[1] Department of Cancer Immunology and Virology, Dana-Farber Cancer Institute, Boston, Massachusetts 02215, USA. [2] Department of Microbiology and Immunobiology, Harvard Medical School, Boston, Massachusetts 02115, USA. [3] Department of Chemical and Biological Engineering, Drexel University, Philadelphia, Pennsylvania 19104, USA. [4] Department of Microbial Pathogenesis, Yale University School of Medicine, New Haven, Connecticut 06536, USA. [5] Department of Chemistry, University of Pennsylvania, Philadelphia, Pennsylvania 19104, USA. [6] Vaccine Research Center, National Institute of Allergy and Infectious Diseases, National Institutes of Health, Bethesda, Maryland 20892, USA. [7] Department of Biology, Johns Hopkins University, Baltimore, Maryland 21218, USA. [8] Department of Physiology and Biophysics, Weill Cornell Medical College of Cornell University, New York, New York 10065, USA. [9] Department of Immunology and Infectious Diseases, Harvard T.H. Chan School of Public Health, Boston, Massachusetts 02115, USA. Correspondence and requests for materials should be addressed to A.H. (email: alon_herschhorn@dfci.harvard.edu) or to J.G.S. (email: joseph_sodroski@dfci.harvard.edu)

The entry of human immunodeficiency virus type 1 (HIV-1) into target cells is mediated by the interaction of viral envelope glycoproteins (Env) with the host CD4 receptor and CCR5/CXCR4 co-receptor[1–7]. There are ~10–14 trimeric Env spikes on the surface of the HIV-1 virion, each composed of three gp120 exterior glycoproteins associated with three gp41 transmembrane glycoproteins. CD4 binding triggers a cascade of conformational changes in HIV-1 Env that result in the transition of the Env from the unliganded, metastable, high-energy state to downstream conformations. These CD4-induced changes involve gp120 (structural rearrangements in the V1/V2 and V3 loops at the trimer apex, formation of a bridging sheet, and exposure of the co-receptor-binding site) and gp41 (formation/exposure of the heptad repeat 1 (HR1) coiled coil)[8–12]. Subsequent engagement of the Env-CD4 complex with the co-receptor (CCR5 or CXCR4) moves the Env down the energy gradient on the entry pathway, culminating in the formation of a gp41 six-helix bundle that facilitates the fusion of viral and cellular membranes[13–17].

Single-molecule fluorescence resonance energy transfer (smFRET) studies provided new insights into the energy landscape of HIV-1 Env and demonstrated that the HIV-1 Env trimer samples three distinct conformational states[18]. Transitions between these states are spontaneous or induced by CD4 binding. These states, designated State 1, State 2, and State 3, have been shown by virological, biochemical, biophysical, and immunologic studies to correspond to the functionally "closed," "intermediate," and "open" Env conformations, respectively[18, 19]. The Envs of primary HIV-1 largely exist in State 1, which is separated by significant activation barriers from States 2 and 3. CD4 binding lowers these barriers and stabilizes States 2 and 3, favoring Env transitions from State 1. Changes in the gp120 V1/V2 region have been shown to release the constraints that maintain State 1, allowing increased occupancy of State 2[19]. These mutant viruses are extremely responsive to CD4 and, compared to the wild-type virus, are hypersensitive to ligands that recognize downstream conformations and are resistant to State 1-preferring ligands[19]. Whether a result of loss of Env restraints or a consequence of CD4 binding, all Env transitions from the functionally "closed" (State 1) conformation to the "open" (State 3) conformation proceed through State 2 as an obligate intermediate[18, 19].

Binding to the CD4 receptor induces allosteric changes in distant domains of the HIV-1 Env trimer through an incompletely understood mechanism[8–12, 18–22]. Structural studies mapped CD4 contacts to a non-continuous set of gp120 residues located at the tip of the β20–β21 hairpin in the bridging sheet, and at the "CD4-binding loop" (α3 helix), $\mathscr{L}$D loop, and β23/β24 strands on the outer domain[23]. CD4 binding induces the rearrangement of the gp120 V1/V2 and V3 regions at the trimer apex and the exposure of the gp41 HR1 coiled coil, Env elements that are distant from the CD4-binding site[10–12, 24]. How CD4 binding induces long-range structural rearrangements in HIV-1 Env is still not well understood. Here, we develop chemical probes and use them together with a variety of molecular techniques, including smFRET and genetic analysis, to study the regulation of HIV-1 transitions upon CD4 binding. We identify the β20–β21 hairpin of gp120 as a site of conformational control in HIV-1 Env, introduce changes in this element that recapitulate the structural rearrangements induced by CD4, and study interactions between β20–β21 and other gp120 elements. The results provide a better understanding of the control of discrete HIV-1 Env transitions to downstream conformations on the virus entry pathway.

## Results

### Rational design identifies chemical probes.
We reasoned that mapping the conserved binding site of chemical probes that affect HIV-1 Env rearrangements during virus entry will assist the identification of key Env residues that regulate conformational transitions. We developed a panel of structurally related compounds, based on an N,N′-difunctionalized piperazine, which is a popular building block for synthesis of chemical libraries and a functional group present in the entry inhibitor BMS-806 (see Methods and Supplementary Tables 1–3). The set of molecules was tested for inhibition of a panel of HIV-1 strains that included transmitted/founder and primary viruses from phylogenetic clades A, B, C, and D. The half-maximal inhibitory concentration ($IC_{50}$) of each compound was determined for each HIV-1 strain (Supplementary Fig. 1). The data were used to cluster the different HIV-1 strains according to their overall sensitivity, and the compounds according to their breadth (Fig. 1a). Notably, the sensitivity profile of the viruses did not segregate with phylogenetic clade, but was specified by strain-dependent determinants (Fig. 1b). Compound **484** exhibited the broadest and most potent anti-HIV-1-specific activity (Fig. 1c) and was further used to study Env conformational transitions.

### Conformational effects of 484 binding.
We used two-color flow cytometry to measure the effects of **484** binding on HIV-1 Env conformation (Supplementary Fig. 2). In the absence of soluble CD4 (sCD4), **484** slightly decreased the binding of the 17b antibody, which recognizes the gp120 bridging sheet[23, 25]. In addition, we observed dose-dependent **484** inhibition of two CD4-induced structural changes: (1) the movement of the V1/V2 region, monitored by the binding of the quaternary antibody PG9, and (2) the exposure of the gp41 HR1 coiled coil, detected with the C34-Ig reagent, which contains the HR2 sequence fused to an immunoglobulin constant region. Thus, **484** impedes CD4-induced Env transitions to downstream conformations that are critical for virus entry[10–12, 24, 26]. Notably, BMS-806 exhibited a more limited effect on CD4-induced Env conformational changes, blocking the exposure of the gp41 HR1 coiled coil but not gp120 V1/V2 movement[24, 26].

We also compared the effects of **484** on HIV-1 Env conformation with those caused by the binding of a previously identified small-molecule CD4-mimetic compound (CD4mc), DMJ-II-121[27]. The effects of DMJ-II-121 binding on Env conformational states completely opposed those observed for **484** binding. DMJ-II-121 increased the exposure of both the gp120 bridging sheet (based upon 17b binding) and the gp41 HR1 coiled coil (based on C34-Ig binding) in a dose-dependent manner (Supplementary Fig. 2). Thus, DMJ-II-121 binding promotes Env transitions from State 1 to States 2 and 3, consistent with its ability to mimic CD4 binding. Conversely, **484** blocks CD4-induced Env transitions from State 1 to downstream conformations.

### Resistance and sensitivity to 484 and DMJ-II-121.
Despite binding to a small region on HIV-1 Env, **484** and DMJ-II-121 modulate large-scale structural rearrangements in the viral spike. We reasoned that information on the binding sites of **484** and DMJ-II-121 could implicate specific regions of HIV-1 Env in the control of transitions between conformational states. We tested a large panel of HIV-1$_{JR-FL}$ variants with single-residue changes in Env for their sensitivity to these compounds. Resistance to **484** resulted from changes in the gp120 β20–β21 element, α1 helix, Phe 43 cavity, and V1/V2 region (Fig. 2a); resistance to DMJ-II-121 was mostly associated with amino acid changes around the gp120 Phe 43 cavity, which constitutes the known binding site for DMJ-II-121 and the other CD4mc[27] (Fig. 2b). A cluster of changes in the V1/V2 region (I154A, N156A, Y177A, and L193A) resulted in viruses that were extremely sensitive to DMJ-II-121

but resistant to **484**. These changes have been shown to destabilize State 1 and increase Env sampling of downstream conformations, indirectly rendering HIV-1 more sensitive to CD4mc and less sensitive to conformational blockers[19, 24]. The resistance-associated changes in gp120 residues Trp 112, Ile 424, Met 426, and Tyr 435 suggest a potential **484**-binding site between the α1 helix and β20–β21 element. Consistent with this interpretation are the significant increases and decreases in **484** sensitivity observed for different substitutions of Met 426, with little effect on sensitivity to DMJ-II-121. Attempts to co-crystallize **484** with

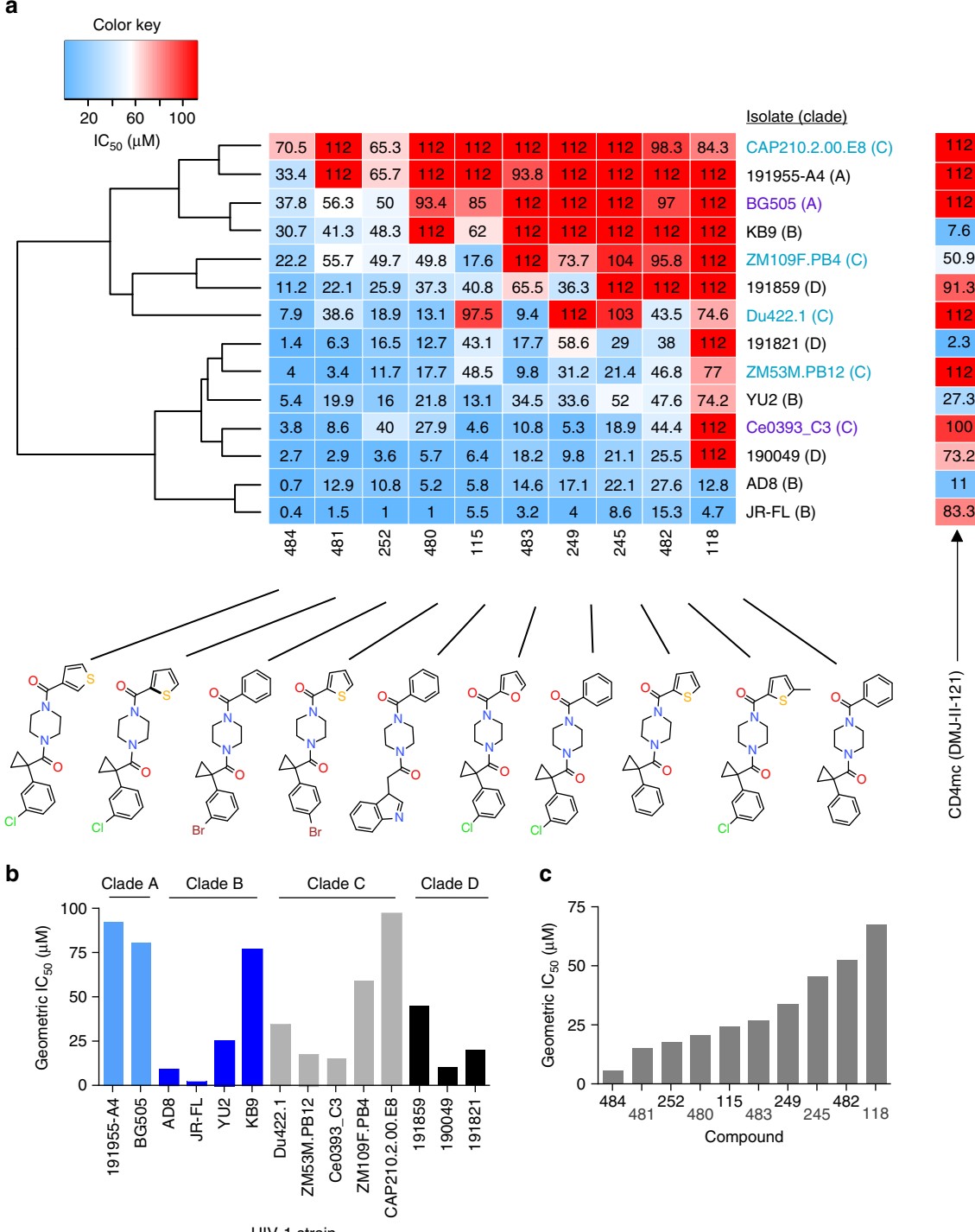

**Fig. 1** Chemical probes of HIV-1 Env function. **a** A panel of chemical probes was developed and tested for inhibition of a diverse set of HIV-1 strains from different clades. The average $IC_{50}$ values were calculated from those obtained in two or three independent experiments. The $IC_{50}$ of each compound for each virus strain is plotted on a heat map; the compounds are ordered according to the geometric mean $IC_{50}$ of each compound against the panel of viruses and the viruses are clustered according to the combination of $IC_{50}$s of the set of compounds against a specific strain. Transmitted/founder, acute/early, and primary isolates are shown with purple, light blue, and black letters, respectively. Under the conditions tested, variation of up to two orders of magnitude in sensitivity to the different compounds was observed across different HIV-1 isolates. **b** The geometric mean $IC_{50}$ of all compounds against each specified HIV-1 strain. **c** The geometric mean $IC_{50}$ of each specified compound against the panel of viruses

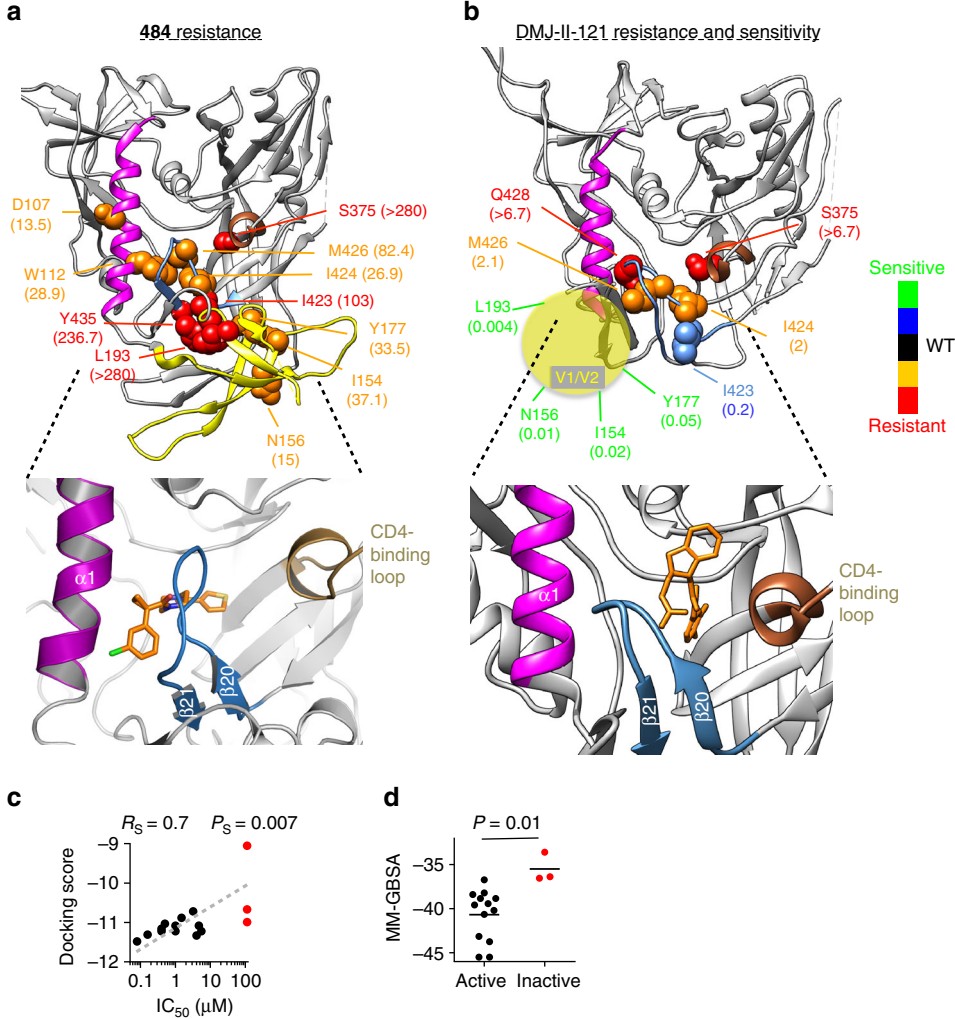

Fig. 2 Genetic analysis and binding sites of chemical probes of HIV-1 Env conformation. **a**, **b** Amino acid residues associated with resistance or hypersensitivity to **484** and the CD4-mimetic compound DMJ-II-121 are shown on structures of the HIV-1$_{BG505}$ soluble gp140 (sgp140) SOSIP.664 glycoprotein. We used an Env structure without sCD4 (Protein Data Bank (PDB) 4TVP)[30] for mapping **484** susceptibility, and a CD4-bound Env conformation (PDB 5THR)[22] for mapping DMJ-II-121 susceptibility. The CD4-bound Env model represents a fit of the sgp140 SOSIP.664 structure to an 8.9-Å-resolution cryo-EM density map; the model lacks the V1/V2 region, which is schematically represented (yellow sphere). In comparison with the structure of sgp140 SOSIP.664 without sCD4, the density map shows a large CD4-induced movement of the V1/V2 region of gp120[22]. The color code key indicates the level of resistance for the specified residues. The ratio of the mutant to wild-type HIV-1$_{JR-FL}$ IC$_{50}$ values (fold change) for resistant and hypersensitive HIV-1 mutants is shown in parentheses; the IC$_{50}$ value of each Env mutant is shown in Supplementary Table 4. Infectivity of the mutant HIV-1$_{JR-FL}$ viruses was not significantly affected by the amino acid changes except for two changes (I154A and N156A). The expanded image in the lower panel of **a** shows a docking pose of the **484** compound in the crystal structure of the HIV-1$_{BG505}$ soluble gp140 SOSIP.664 component of the complex with BMS-626529[28]. The expanded image in the lower panel of **b** shows the crystal structure of DMJ-II-121 in complex with the HIV-1$_{C1086}$ gp120 core (PDB ID 4I53).[27] **c**, **d** The relationship between either the docking scores (**c**) or docking energy (**d**) and the antiviral activity of a panel of **484** derivatives was used to assess the reliability of the docking in **a**. The dashed line indicates the linear trend between the docking score and IC$_{50}$ values. The scores and derivatives analyzed are shown in Supplementary Table 5. As a control, the BMS-626529 compound was docked into the crystal structure (Supplementary Fig. 4). Red color, non-active compounds (IC$_{50}$ > 112 μM). $R_S$, Spearman coefficient; $P_S$, Spearman P value; P, t test P value

the HIV-1$_{BG505}$ soluble gp140 (sgp140) SOSIP.664 Env were unsuccessful, but parallel efforts determined the structure of other conformational blockers, BMS-806 and BMS-626529, in a complex with this Env trimer[28]. We used the crystal structure of HIV-1$_{BG505}$ sgp140 SOSIP.664-BMS-626529 (after removing the bound BMS-626529) to model the binding of **484**. Docking several newly designed and synthesized **484** analogs (i.e., **484-3**–**484-18** in Supplementary Table 5) into the Env trimer structure led to binding scores that correlated with the IC$_{50}$s of the compounds, emphasizing the reliability of this analysis and indicating a binding site close to the β20–β21 element (Fig. 2c, d, Supplementary Table 5, and Supplementary Figs. 3–5). The proposed

binding site was consistent with the ability of **484** to decrease the binding of the 17b antibody, which contacts β20–β21, as well as the location of gp120 changes associated with **484** resistance. Notably, this model suggests that CD4mc and conformational blockers like **484** bind proximal to and on opposite sides of the gp120 β20–β21 element, which also contains residues that contact CD4[23, 29, 30].

**Involvement of gp120 β20–β21 in maintaining Env State 1.** Mapping the binding sites of CD4mc and conformational blockers to opposite sides of the gp120 β20–β21 structure

suggested that residues within this gp120 element might control Env structural rearrangements. Substantial repositioning of β20–β21 after sCD4 binding further supports this hypothesis (Fig. 2a, b). We therefore used Env ligands that recognize downstream conformations to evaluate the effect of changes in the gp120 β20–β21 structure on the conformational state of Env. These ligands included sCD4; the 19b monoclonal antibody (Mab) directed against the gp120 V3 loop; MAb 902090 and Fab

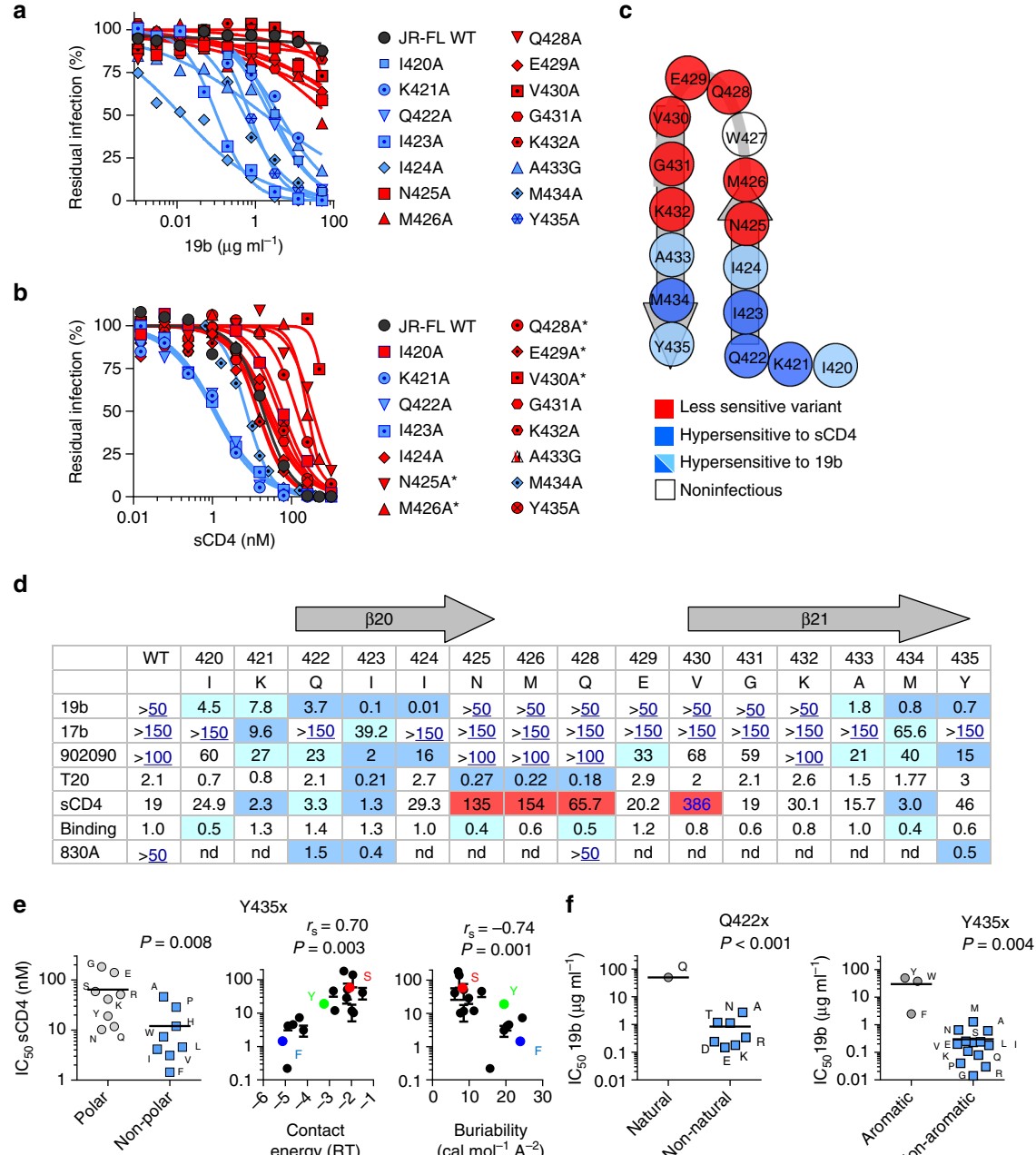

**Fig. 3** Involvement of the gp120 β20–β21 element in regulation of HIV-1 Env conformational transitions. **a**, **b** Effect of single-residue changes in the gp120 β20–β21 hairpin on the sensitivity of HIV-1$_{JR-FL}$ to neutralization by the V3-directed 19b antibody **a** and by sCD4 **b**. Changes that increased HIV-1 susceptibility to the specified ligand are shown in blue and all others in red. Residues that contact CD4 are indicated with an asterisk. **c** Phenotypes associated with gp120 β20-β21 residues. Trp 427 could not be tested due to the low level of replication of the W427A and W427F viruses. **d** Average IC$_{50}$ values of inhibition of HIV-1$_{JR-FL}$ with the β20–β21 changes listed in **a** and **b** by conformation-sensitive Env ligands. Reported units are μg ml$^{-1}$ for 19b, 17b, 902090, and 830 A, and nM for sCD4 and T20; sCD4 binding to the cell-surface Env is normalized to the WT Env values. Reported values for sCD4 inhibition were normalized for sCD4 binding. When IC$_{50}$ values were above the tested concentrations, the highest concentration tested is shown in blue letters and is underlined. Values that were significantly below or above the ones obtained for WT HIV-1$_{JR-FL}$ are highlighted with blue and red backgrounds, respectively. **e** Relationships between the effect of changes in residue 435 on the sensitivity of HIV-1$_{JR-FL}$ to sCD4 and the polarity, contact energy (in RT units, $R$ = universal gas constant and $T$ = temperature)[48], and buriability[49] for each amino acid change. **f** The effect of changes in residue 422 (left) and residue 435 (right) on the sensitivity of HIV-1$_{JR-FL}$ to the V3-directed 19b antibody. $P$ values were calculated using a one-sample $t$ test (**f**, left), a Mann–Whitney test for the difference between the groups (**e**, left and **f**, right), or Spearman correlation (**e**, middle and right). Results shown are the average of those obtained in two or three independent experiments (see also Supplementary Fig. 7). WT, wild-type

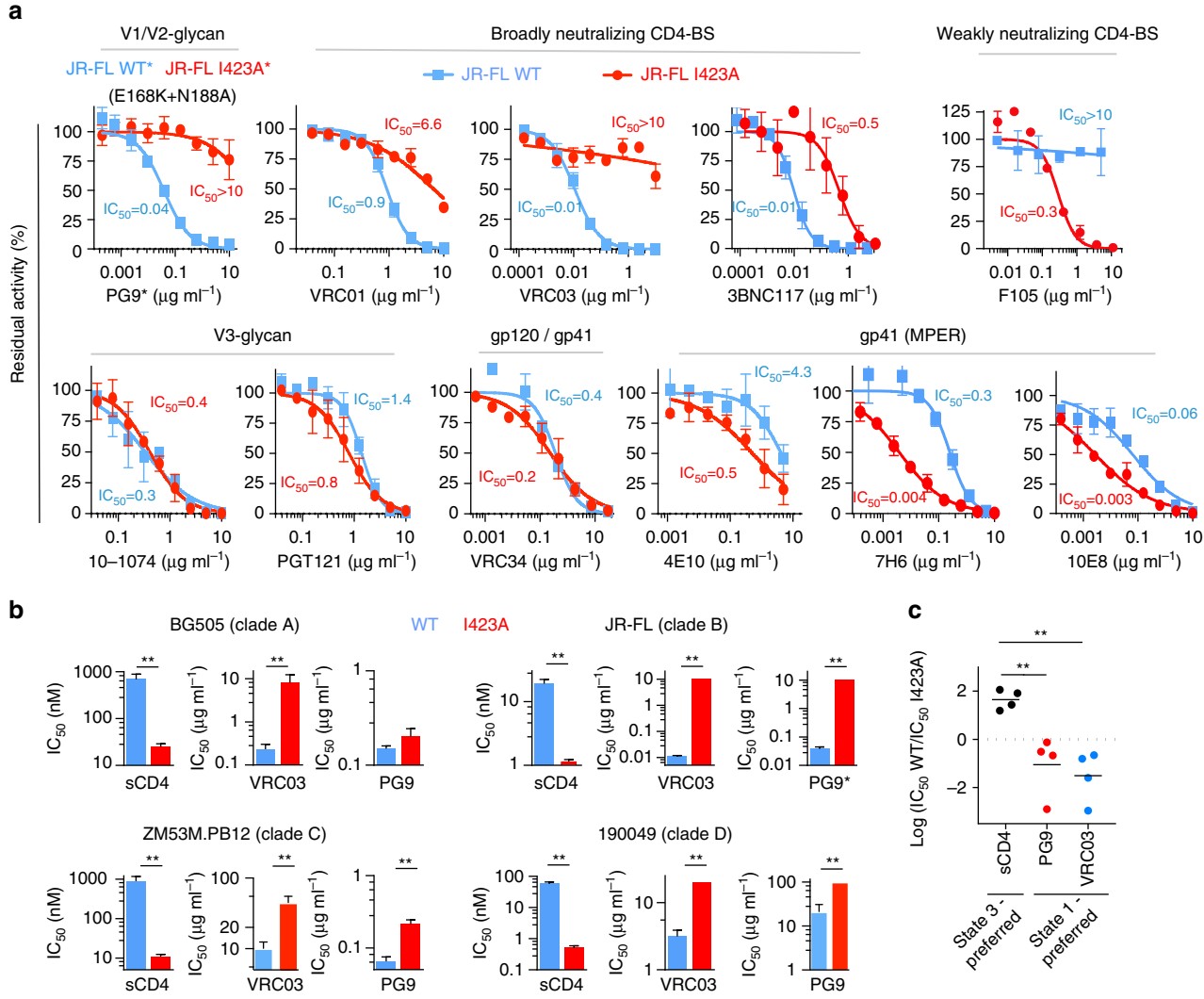

**Fig. 4** The effects of the I423A change in different primary HIV-1 strains on Env conformational state and sensitivity to broadly neutralizing antibodies. **a** Neutralization of wild-type (WT) (blue) and I423A (red) HIV-1$_{JR-FL}$ by different bNAbs. The calculated IC$_{50}$s are shown. **b** Conformation-selective bNAbs and sCD4 were used to test the sensitivity of the WT (blue) and the I423A variant (red) of HIV-1 strains from clades A, B, C, and D. Reported IC$_{50}$ units are nM for sCD4 and µg ml$^{-1}$ for the bNAbs. The asterisk (*) indicates that sensitivity of the PG9 antibody was tested against the HIV-1$_{JR-FL}$ E168K + N188A Env (with and without the I423A change); these V2 changes restore the PG9 epitope in the HIV-1$_{JR-FL}$ Env[32, 47]. **c** Comparison of the sensitivity of all isolates from **b** to ligands that prefer State 1 and State 3. Results shown are the average of those obtained in two or three independent experiments and error bars represent s.e.m. The double asterisk (**) indicates one-tail *P* values <0.05

830A, directed against a V2 β-barrel; 17b, a CD4-induced (CD4i) MAb; and T-20, an HR2 peptide that targets the gp41 HR1 coiled coil (Supplementary Table 6). Specific changes in the base of the β20–β21 structure (residues 421–424 and 433–435) resulted in HIV-1$_{JR-FL}$ hypersensitivity to the ligands recognizing downstream (State 2 and State 3) Env conformations (Fig. 3). Of note, the I423A mutant was hypersensitive to multiple ligands that recognize distinct Env structural elements involved in conformational transitions: V3, V1/V2, and β20–β21 (in gp120) and HR1 (in gp41) (Fig. 3a–d). The I423A virus exhibited increased sensitivity to State 2-/State 3-recognizing ligands, such as the weakly neutralizing antibody against the CD4-binding site (F105) and several MPER-directed antibodies (4E10, 7H6, and 10E8), which have been shown to recognize downstream (State 2/State 3) Env conformations preferentially[19, 24]. By contrast, the I423A virus was relatively resistant to neutralization by broadly neutralizing antibodies that have been shown to bind State 1 Env preferentially (i.e., the PG9 antibody and the CD4-binding site antibodies VRC01, VRC03, and 3BNC117

(Fig. 4a, b))[18, 19, 24, 31–33]. These results indicate that, relative to wild-type HIV-1$_{JR-FL}$, the I423A virus more frequently samples States 2 and 3.

Virus sensitivity to sCD4 could be modulated by altering the buriability of the highly conserved Tyr 435, which does not contact CD4[23] (Fig. 3e). The presence of a hydroxyl group at residue 435 on the background of different isolates may be important in vivo, as most HIV-1 isolates that contain a residue other than tyrosine have a serine residue at this position (Supplementary Fig. 6). Almost all changes in Tyr 435 and Gln 422 led to increased virus sensitivity to the 19b anti-V3 loop antibody (Fig. 3f and Supplementary Fig. 7); in some Env structures (PDB 4TVP), these residues are close to each other and to the V3 region. Overall, the β20–β21 gp120 region contacts CD4 and contains residues that are involved in the maintenance of State 1. Specific alterations of these residues modulate allosteric changes in the trimer apex (V1/V2 and V3 regions), the gp41 ectodomain, and the CCR5-binding site.

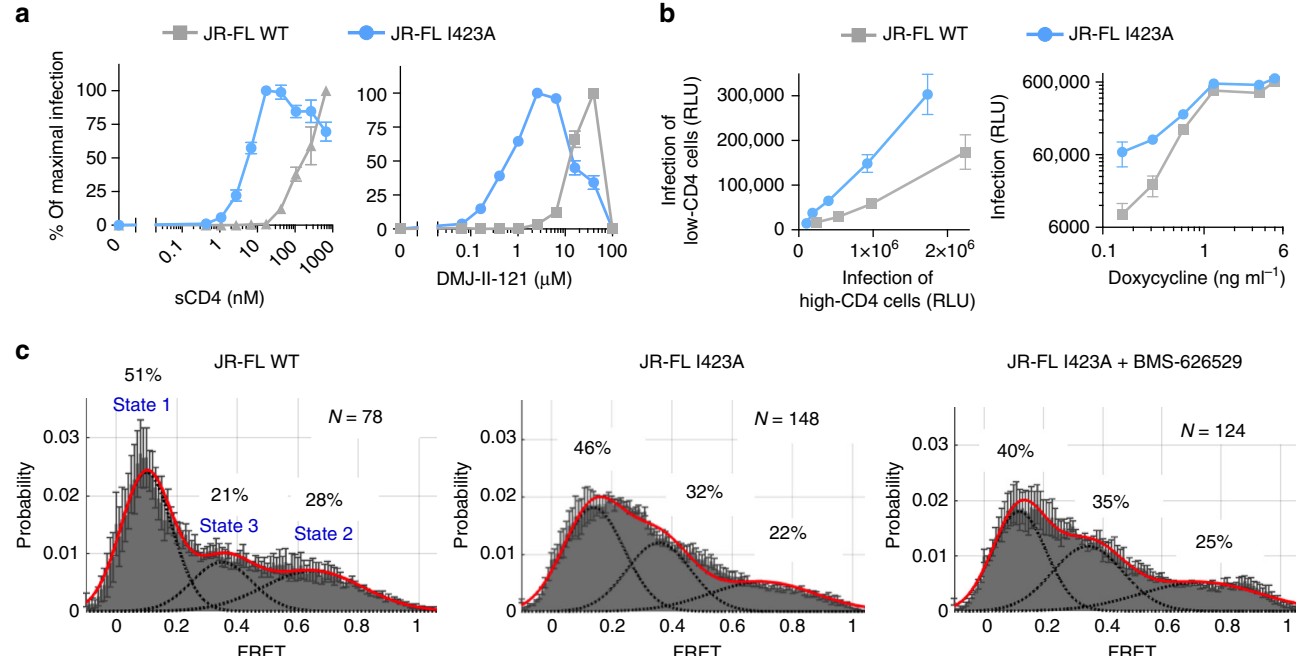

**Fig. 5** The effect of the I423A change on CD4 dependence and Env conformation. **a** Recombinant viruses carrying the wild-type (WT) or I423A HIV-1$_{JR-FL}$Env were incubated with CD4-negative, CCR5-expressing Cf2Th cells in the presence of the CD4-mimetic compound DMJ-II-121 or sCD4 at the indicated concentrations. The percentage of maximal infection for each virus variant is reported here. The maximal infection levels after DMJ-II-121 addition were 297,656 and 70,626 relative luciferase units for viruses with the WT and I423A Envs, respectively. Error bars represent s.e.m. **b** Infection of cells with different levels of CD4. Left, parallel titration of wild-type (WT) or I423A HIV-1$_{JR-FL}$ on affinofile cells that were induced to express low or high levels of CD4. Right, the effect of different levels of CD4 expression, induced by doxycycline treatment of affinofile cells, on the infectivity of WT or I423A HIV-1$_{JR-FL}$. Maximal infection levels were 675,978 and 613,708 relative luciferase units for viruses with the WT and I423A Envs, respectively. **c** Single-molecule fluorescence resonance energy transfer (smFRET) probes were placed in the gp120 V1 and V4 loops of WT or I423A HIV-1$_{JR-FL}$ Envs. The FRET trajectories were compiled into population FRET histograms and fit to the Gaussian distributions associated with each conformational state, according to a hidden Markov model[18]. The percentage of the population that occupies each state as well as the number of molecules analyzed is shown and represents the average of two independent experiments

**Conserved effect of the I423A change on Env conformation.** We used broadly reactive Env ligands that preferentially recognize specific Env conformations to test the effects of the I423A change in different HIV-1 isolates. Virus sensitivity to sCD4 (State 2/3-preferring) and to the PG9 and VRC03 (State 1-preferring) antibodies was used as a surrogate for Env conformational transitions[19, 32, 33]. The I423A mutants from all HIV-1 clades tested were more sensitive to sCD4 inhibition and more resistant to PG9 and VRC03 inhibition in comparison to the related wild-type viruses (Fig. 4c). The general effect of the I423A change in different HIV-1 strains is consistent with the high degree of conservation of the amino acids that form the β20–β21 element, including Ile 423 (90.5% among 2500 primary HIV-1 isolates). Moreover, substitutions that naturally occur at reside 423 are mostly limited to amino acids that maintain State 1 (Supplementary Fig. 8). These observations support the important role of Ile 423 in maintaining a State 1 conformation in different HIV-1 strains.

**β20–β21 changes decrease CD4 requirements for HIV infection.** Since CD4-induced Env transitions to downstream conformations can be recapitulated by changes in β20–β21 such as I423A, we measured the CD4 requirement of viruses carrying this change (Fig. 5a, b). The I423A mutant required significantly less CD4mc or sCD4 to trigger virus entry than did the wild-type Env. The I423A mutant also infected cells that express low levels of CD4 more efficiently than the wild-type virus (Fig. 5b). These results indicate that, compared with the wild-type HIV-1$_{JR-FL}$, the I423A

mutant needs less CD4 to make the transition into the CD4-bound conformation.

To examine the conformational states of the I423A mutant directly, we used smFRET analysis to study the I423A Env in the presence and absence of a conformational blocker, BMS-626529 (Fig. 5c). This analysis showed that, compared to the wild-type Env, the I423A mutant exhibited decreased occupancy of State 1 and increased occupancy of State 3. Conformational blockers like BMS-626529 have been shown to decrease HIV-1 Env transitions from State 1, leading to increased occupancy of State 1[18, 19, 24]. The distribution of the I423A conformational states was minimally affected by BMS-626529 treatment. The relative increase in the spontaneous sampling of downstream conformations by the I423A mutant explains the sensitivity of this virus to Env ligands that preferentially bind these conformations.

**Interactions between the gp120 β20–β21 and V1/V2 regions.** We recently reported that Leu 193 in the gp120 V1/V2 region helps to maintain Env from diverse HIV-1 strains in State 1[19]. Given the similarities in the HIV-1 phenotypes associated with changes in the gp120 V1/V2 and β20–β21 regions, we investigated potential functional interactions between these gp120 elements. The phenotypes of HIV-1$_{JR-FL}$ mutants with alterations in either Leu 193 or Ile 423 were compared with mutants containing changes in both residues. Both the L193A and I423A mutants exhibited dramatic increases in sensitivity to sCD4, the 19b anti-V3 antibody, and the 902090 anti-V2 antibody, consistent with the expected movement of these mutants from State 1 to

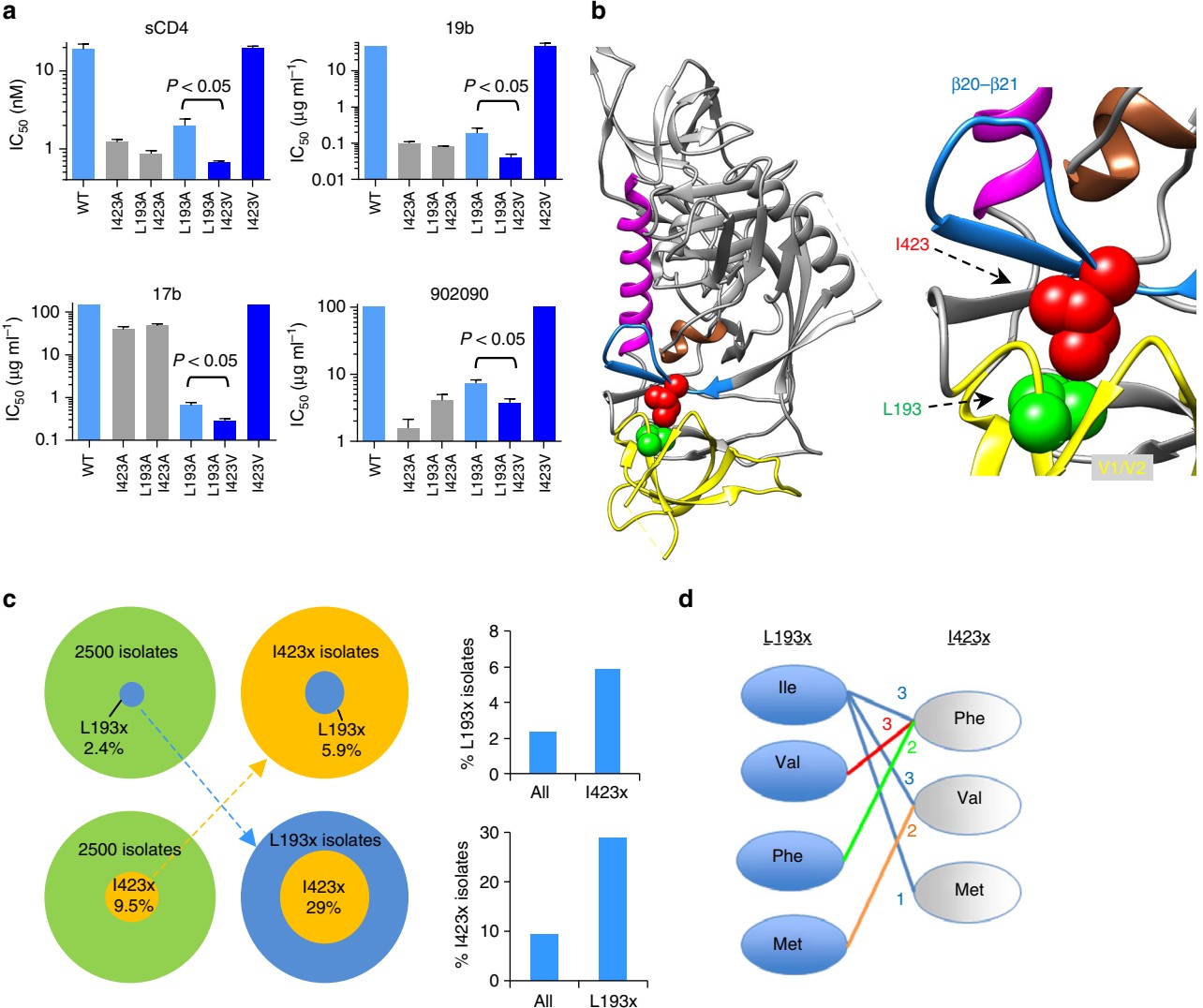

**Fig. 6** Interaction between residues in the gp120 β20–β21 element and the V1/V2 region. **a** The individual and combined effect of changes in Ile 423 and Leu 193 on the sensitivity of HIV-1 to ligands recognizing downstream conformations. Results shown are averages of those obtained in two or three independent experiments and error bars represent s.e.m. Indicated *P* values were calculated using a two-sample *t* test. **b** Leu 193 and Ile 423 were mapped on a structure of HIV-1 Env bound to the PGT151 antibody (PDB ID 5FUU)[36]. **c** Analysis of the prevalence of amino acids other than isoleucine at position 423 or leucine at position 193 among 2500 primary HIV-1 strains. Green pie plots show the prevalence in all HIV-1 strains and residue-specific pie plots (set to the same size as the green plots) show the prevalence of specific amino acids within the HIV-1 subpopulation that carries amino acids other than isoleucine at position 423 or leucine at position 193. **d** Possible combinations of different amino acids at Env residues 193 and 423 in primary HIV-1 strains. The number of isolates with the specified pairing is indicated

downstream conformations (Fig. 6a). The difference in the level of sensitivity of these two mutants to the 17b CD4i antibody likely results from the partial disruption of the 17b epitope by the I423A change[34], which was also manifested in the low 17b sensitivity of the L193A/I423A mutant. The sensitivity of the L193A/I423A mutant to sCD4, 19b, and 902090 was similar to those of the Env mutants with the individual residue changes. Thus, the phenotypic effects of the L193A and I423A changes on HIV-1$_{JR-FL}$ sensitivity to sCD4 and anti-V2/V3 antibodies are redundant.

We next examined the phenotypes associated with the I423V change in the contexts of wild-type HIV-1$_{JR-FL}$ or an L193A mutant. The change in isoleucine 423 to valine has been previously reported to contribute to the CD4-independent phenotype of the laboratory-adapted HIV-1$_{HXBc2}$, but showed no effect on HIV-1$_{JR-FL}$[35]. Consistent with these results, we found no significant effect of this change on HIV-1$_{JR-FL}$ sensitivity to sCD4 or the 17b, 19b, and 902090 antibodies (Fig. 6a). However,

the addition of the I423V change to the L193A mutant virus significantly enhanced its sensitivity to these different ligands, which recognize downstream Env conformations. Thus, the effect of the I423V change on the conformational state of Env is dependent upon the presence of the L193A change. These observations suggest a model in which the L193A and I423A changes release the restraints on State 1, allowing Env to populate the downstream States 2 and/or 3. On its own, the I423V change does not appreciably destabilize State 1; however, once State 1 has been destabilized by the L193A change, the I423V change may facilitate transitions between downstream states (e.g., between State 2 and State 3).

The above mutagenesis study indicates functional cooperativity between two different gp120 residues: Leu 193 in the V1/V2 region and Ile 423 in the β20/β21 element. Of note, Leu 193 and Ile 423 are in close proximity on some structures of HIV-1 Env complexed with neutralizing antibodies[30, 36], suggesting a

**Table 1 Frequency of non-consensus amino acids in each of the β20–β21 residues among 2500 primary HIV-1 strains (All) and among those HIV-1 strains that contain a residue other than leucine at position 193 (L193x)**

| Residue | All (%) | L193x (%) | Enrichment[a] | Residue | All (%) | L193x (%) | Enrichment |
|---|---|---|---|---|---|---|---|
| I420x | 1 | 2 | 2 | E429x | 55 | 39 | 0.7 |
| K421x | 8 | 6 | 0.75 | V430x | 10 | 12 | 1.2 |
| Q422x | 1 | 2 | 2 | G431x | 1 | 2 | 2 |
| I423x[b] | **9.5** | **29** | **3.1** | K432x | 61 | 43 | 0.7 |
| I424x | 25 | 33 | 1.3 | A433x | 2 | 2 | 1 |
| N425x | 9 | 14 | 1.6 | M434x | 12 | 6 | 0.5 |
| M426x | 16 | 27 | 1.7 | Y435x | 1 | 0 | 0 |
| W427x | 1 | 0 | 0 | | | | |
| Q428x | 2 | 2 | 1 | Average | | | 1.2 |

[a]Enrichment, ratio of the frequency among non-leucine 193 HIV-1 strains to the frequency among all HIV-1 strains. The values associated with the maximal enrichment are indicated in bold.
[b]P value < 0.00001 in a two-tailed *t* test for the difference between the enrichment of I423x and other residues

possible mechanism for the cooperative regulation of Env transitions (Fig. 6b). These observations prompted us to analyze the covariation of these two residues in different primary HIV-1 strains. Analysis of 2500 *env* sequences (from the HIV-1 database, https://www.hiv.lanl.gov/) showed that Leu 193 and Ile 423 are both highly conserved, with 97.6 and 90.5% identity, respectively, among different HIV-1 strains. Remarkably, HIV-1 strains that contain a residue other than isoleucine at position 423 exhibit a 2.5-fold enrichment in substitutions for leucine at position 193. Similarly, a change in Leu 193 was associated with a 3.1-fold increase in substitutions for isoleucine at position 423 (Fig. 6c and Table 1). Analyzing this effect in each of the β20–β21 residues showed that the degree of co-variation with Leu 193 was greater for Ile 423 than for any other β20–β21 residue (Table 1). Thus, optimal Env configurations may require compatible combinations of residues at these positions. The identification of less prevalent HIV-1 strains that have Env residues other than leucine (at position 193) and isoleucine (at position 423) allowed us to assess the compatible combinations of these residues on the background of different isolates (Fig. 6d). An isoleucine residue at 193 was paired with different hydrophobic amino acids at position 423 and this apparent flexibility is consistent with the increased ability of Envs with Ile 193 to maintain State 1, relative to Envs with Leu 193[19]. Other substitutions at position 193 were less tolerated and each matched a specific residue at position 423, indicating that in diverse HIV-1 strains, there may be subtly different requirements for maintaining β20–β21–V1/V2 interactions.

## Discussion

We developed chemical probes of HIV-1 Env function that block two CD4-induced conformational changes and used them to map a control switch for Env structural rearrangements. Compound **484** exhibits broad range activity against diverse HIV-1 strains from different clades and interferes with Env structural rearrangements, making it an attractive probe for mapping conserved regulatory elements in HIV-1 Env. HIV-1 Env changes that resulted in resistance to **484** inhibition were of two types. One set of changes, in the gp120 V1 and V2 variable regions and the β20–β21 element, resulted in HIV-1 resistance to **484** and increased sensitivity to a CD4-mimetic compound. This phenotype is associated with Env changes that destabilize State 1 and increase Env transitions from State 1 to downstream conformations[19, 24]. Recently, changes in HIV-1 gp41 have been described that stabilize State 1 and render the virus more sensitive to inhibition by **484** and more resistant to CD4-mimetic compounds[37]. Thus, HIV-1 variants with higher occupancy of State 1 often exhibit increased sensitivity to inhibition by conformational blockers like **484**. The second set of gp120 changes that conferred

**484** resistance defines a conserved hydrophobic pocket flanking the gp120 β20–β21 element and the α1 helix. Several changes in the β20–β21 element dramatically reduced HIV-1 sensitivity to **484** without global effects on virus susceptibility to conformation-sensitive Env ligands; these results are most consistent with direct disruption of the **484**-binding site. This assertion is further supported by modeling, which takes advantage of very recent co-crystal structures of other conformational blockers, BMS-626529 and BMS-806[28], and accurately predicts the antiviral potency of **484** analogs. The proposed binding site is consistent with the **484**-mediated decrease in the binding of the 17b antibody, which recognizes the gp120 bridging sheet formed upon CD4 binding[23, 25, 34]. In the CD4-bound gp120 structure[23], the residues implicated in **484**-binding line an internal water-filled channel, explaining the poor accessibility of the binding site in State 3 to conformational blockers like **484**[38].

Mapping the potential **484** binding site allowed us to identify residues in the gp120 β20–β21 element important for the regulation of conformational changes of the HIV-1 Env. Alteration of these key residues in the base of the β20–β21 β-hairpin recapitulated several conformational changes induced by CD4 binding. For example, alteration of Ile 423 to alanine resulted in a decreased Env occupancy of State 1 and increased spontaneous sampling of the CD4-bound state (State 3). The I423A mutant is resistant to Env ligands that prefer State 1 (conformational blockers, some bNAbs) and hypersensitive to Env ligands that prefer downstream conformations (sCD4, CD4-mimetic compounds, and some antibodies). The I423A virus requires fewer CD4 molecules to infect cells, although it does not become completely CD4-independent.

The metastable HIV-1 Env trimer is maintained in State 1 by multiple intersubunit and intramolecular interactions[19, 37, 39, 40, 41]. Alteration of key restraining residues destabilizes State 1 and releases the Env trimer to sample downstream conformations. The location of restraining residues identified in this and a previous study[19] suggests a potential mechanism for the induction of structural rearrangements by the CD4 receptor (Fig. 7). According to this model, CD4 contacts with the loop connecting the β20 and β21 strands[23] disrupt interactions in the base of the β20–β21 hairpin that stabilize State 1. The binding of conformational blockers (**484**, BMS-806, BMS-626529) in the adjacent hydrophobic gp120 pocket prevents this destabilizing disruption. Interestingly, peptides derived from the β19–β20 region form nanofibrils in solution, suggesting that out of the gp120 context this region can adopt alternative conformations[42] (Supplementary Fig. 9). The base of the β20–β21 element is proximal to the base of the V3 region, which, along with V1/V2, forms the Env trimer apex in all available structures[20–22, 30, 36]. In some Env structures, Leu 193 constitutes part of the hydrophobic

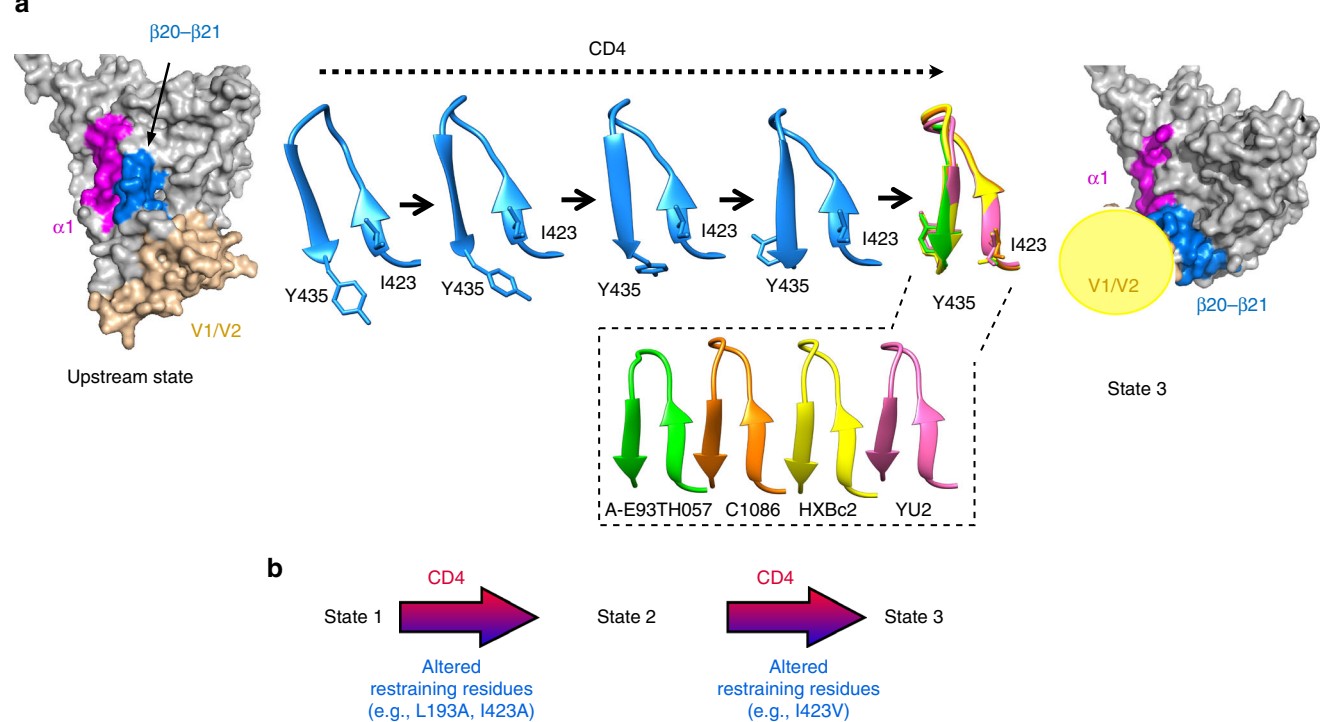

**Fig. 7** Model of HIV-1 Env conformational regulation. **a** Changes in the β20–β21 conformation upon CD4 binding. Left, surface representation showing the location of the β20–β21 element in one gp120 subunit on the HIV-1 Env structure; the ribbon structure of β20–β21 is depicted to the right of the Env surface. Both representations are derived from the crystal structure of the HIV-1$_{BG505}$ sgp140 SOSIP.664 glycoprotein (PDB ID 4TVP)[30]. Right, surface representation from the cryo-EM structure of HIV-1$_{BG505}$ sgp140 SOSIP.664 bound to sCD4 (PDB 5THR; the V1/V2 region is shown schematically as a yellow sphere). The β20–β21 elements from four crystal structures of gp120 from different HIV-1 clades bound to sCD4 or the DMJ-II-121 CD4-mimetic compound (PDB IDs 4I53, 4I54, 1GC1, and 1RZK) are aligned. A possible trajectory between the upstream state and the CD4-bound state was generated with the program Chimera[50]. **b** Effects of CD4 binding on Env conformation. CD4 contacts the gp120 β20–β21 element, altering the conformation of the β20–β21 base. Both CD4 binding and changes in restraining residues allow Env to make the transition from State 1 to downstream conformations. Examples are shown of changes in restraining gp120 residues that affect particular Env transitions

core of a V2 β-barrel and is proximal to Ile 423 in the β20–β21 hairpin[30, 36, 43, 44]. These observations suggest a possible mechanism for conformational regulation of Env transitions by CD4. CD4-induced changes in the β20–β21 base may destabilize the hydrophobic core that contains Leu 193, a restraining residue in the V2 region, and lead to opening the trimer apex. Notably, the β20–β21 element is the only single gp120 component that contacts CD4, is proximal to the V3 and V1/V2 loops, and forms part of the co-receptor-binding site. This may allow β20–β21 to coordinate CD4 binding with transducing a signal for structural rearrangements to distant regions.

Our study provides insights into the allosteric regulation of HIV-1 Env conformational changes by CD4. These insights will assist the design of inhibitors and the stabilization of specific Env conformations for use as immunogens and in structural studies.

## Methods

**Cells.** Cf2Th and 293T were from the American Type Culture Collection. Stocks were tested for mycoplasma using the MycoAlert detection assay (Lonza).

**Identification of new chemical probes.** We virtually screened a chemical database (Enamine) and identified 20 compounds with diverse chemical groups connected by the selected diketo-piperazine scaffold. These molecules were tested for virus inhibition and three compounds specifically inhibited HIV-1 entry. The most potent compound, **118**, was used as a scaffold for sequential screening in two cycles of selection. Follow-up derivatives were subsequently tested for specific HIV-1 inhibition (Supplementary Tables 1–3). The iterative process resulted in a panel of chemical probes with diverse properties and virus-inhibitory potency.

**Compounds.** Most compounds were purchased from Enamine. A few compounds were synthesized, using the synthetic procedures detailed in the Chemical synthesis section of the Supplementary Methods. All compounds were >90% pure.

**Site-directed mutagenesis.** Mutations were introduced into the plasmids expressing full-length HIV-1$_{YU2}$ or HIV-1$_{JR-FL}$ Envs with the QuikChange II site-directed mutagenesis protocol or the QuikChange multi site-directed mutagenesis kit (Stratagene). We confirmed the existence of the mutations by DNA sequencing. Residues are numbered based on alignments with the HXBc2 prototypic sequence, according to convention.

**Virus production.** The 293T cells were co-transfected with an HIV-1 Env-expressing plasmid, pHIVec2.luc plasmid, and psPAX2 plasmid (cat# 11348, NIH AIDS Research and Reference Program) at a ratio of 1:6:3 using Effectene (Qiagen). The supernatant was collected after 48 h, buffered with 50 mM HEPES (final concentration) and centrifuged for 5 min at 2000 r.p.m. and 4 °C. The virus-containing supernatant was used directly or frozen at −80 °C. The amount of p24 in the supernatant was measured using an HIV-1 p24 antigen capture assay (cat# 5421, Advanced BioScience Laboratories).

**Viral infection assay.** The viral infection assay was done as previously described[24]. Briefly, each test compound was serially diluted in a 96-well B&W isoplate (Per-kinElmer) using HP D300 Digital Dispenser. DMSO was used as a control and the final volume of either diluted compound or DMSO was 450 nl. Supernatant (15 ng p24 Gag) containing viruses pseudotyped with a specific Env was added to each well and incubated briefly at room temperature. For poorly infectious viruses, the maximum volume within the assay limit was used. Cf2Th-CD4/CCR5 cells (derived from Cf2Th cells) were detached using the StemProAccutase Cell Dissociation Reagent (Invitrogen, cat# A11105-01), washed once, and 50 μl of $1 \times 10^5$ cells per ml was added to each well. Following a 48-h incubation, the medium was aspirated and cells were lysed with 30 μl of Passive Lysis Buffer (Promega, cat#E1941). Activity of the firefly luciferase, which served as a reporter protein in the system, was measured with a Centro LB 960 luminometer (Berthold

Technologies, Tennessee, USA) as previously described[45]. In some experiments, the compounds (in DMSO) or antibodies (in DMEM medium) were added manually. All inhibition curves were fitted to the four-parameter logistic equation; $IC_{50}$ values and the associated s.e. are reported[46].

**Flow cytometry.** Plasmids expressing wild-type HIV-1$_{JR-FL}$Env$\Delta$CT or the double-mutant HIV-1$_{JR-FL}$Env$\Delta$CT E168K/N188A, which allows recognition by some quaternary V1/V2 antibodies[32, 47], were used for the flow cytometric studies. Plasmids were transfected using the Effectene transfection reagent (Qiagen) into 293T cells, according to the manufacturer's instructions. Cells were detached after 48–72 h with 5 mM EDTA/PBS, and about 0.5–1 million cells were briefly incubated with various concentrations of a test compound and then with or without the indicated concentrations of sCD4. C34-Ig (at a final concentration of 20 µg ml$^{-1}$) or a specified antibody (at a final concentration of 1 µg ml$^{-1}$) was added to the cells. Following a 30-min incubation, cells were washed twice and incubated with allophycocyanin-conjugated F(ab')$_2$ fragment donkey anti-human IgG antibody (1:100 dilution; catalog no. 709-136-149; Jackson ImmunoResearch Laboratories) and fluorescein isothiocyanate-conjugated anti-CD4 antibody (1:33 dilution; catalog no. 11-0048-42; E-biosciences) for 15 min. Cells were washed twice and analyzed with a BD FACSCanto II flow cytometer (BD Biosciences).

**Molecular modeling.** Target and ligand preparation: The X-ray co-crystal structure of the BMS-626529 complex with the HIV-1$_{BG505}$ soluble gp140 SOSIP.664 Env trimer[28] was prepared for docking by the Protein Preparation Wizard software of the Schrödinger 2016-4 suite. This included adding H atoms, assigning bond orders, filling in missing side chains, and checking amino acid protonation states at physiological pH using PROPKA and the co-crystallized ligand ionization state at the same pH using EPIK. Compounds were designed with Maestro 11 and pre-treated using LigPrep (Schrödinger 2016-4 suite). Their ionization state at physiological pH was analyzed using EPIK and their tautomers were generated.

Docking: Rigid receptor docking was performed using Glide 7.2 (Schrödinger 2016-4 suite). The grid was built on the pretreated target protein and centered on the co-crystallized BMS-626529, with an inner box size of $10 \times 10 \times 10$ Å and outer box size of $30 \times 30 \times 30$ Å. Pretreated compounds were docked using the standard precision (SP) scoring function with the number of poses that undergo post-docking energy minimization set to 50.

Rescoring: Docking results were rescored by computing the protein–ligand interaction energy using the Molecular Mechanics/Generalized Born Surface Area (MM-GBSA) method on a 10 ns molecular dynamics (MD) simulation in water as explicit solvent by means of NAMD 2.10 and the CHARMM26 force field. The MD simulation was validated on the BMS-626529/sgp140 SOSIP.664 co-crystal coordinates[28].

**Statistical analysis.** Statistical parametric analyses were performed by unpaired Student's $t$ tests. Statistical nonparametric analyses were done by Spearman rank correlation and Mann–Whitney tests. We used the nonparametric Levene test to compare the variance within the different groups that were tested by the Mann–Whitney test; no significant difference was found between the variance of the polar and non-polar groups in Fig. 3e ($P$ value >0.05) and a significant difference was found between the variance of the aromatic and non-aromatic groups in Fig. 3f ($P$ value <0.05). Summary of the data statistics is included for each analysis in the relevant figure legend.

**Sequence analysis of HIV-1 isolates.** The online tool on the HIV-1 database website (https://www.hiv.lanl.gov/) was used to retrieve the DNA codon at specific positions in the HIV-1 genome. The data were exported to an Excel worksheet and the frequency of the translated amino acid was compared among all HIV-1 strains. The complete Env sequences of HIV-1 strains with amino acids other than leucine at position 193 were retrieved, exported, and aligned using Clustal omega (http://www.ebi.ac.uk/Tools/msa/clustalo/). The amino acids around Env position 193 and the residues comprising the complete β20–β21 element were compared among HIV-1 strains (Supplementary Table 7).

**Data availability.** Relevant data are available from the corresponding authors upon reasonable request.

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

## Acknowledgements

We thank E. Carpelan for manuscript preparation; the AIDS Research and the Reference Reagent Program, Division of AIDS, NIAID, NIH for providing the following anti-HIV-1 Env antibodies: VRC01, VRC03, 3BNC117, F105, 10-1074, 10E8, 4E10, 7H6, and 35O22, the psPAX2 plasmid and T20. We also thank Drs. D. Easterhoff, T. Bradley, and B. Haynes for providing the 902090 expression plasmids, Dr. J. Robinson (Tulane University) for the 17b and 19b expression plasmids, Dr. J. Mascola for the VRC34 expression plasmids and Dr. X.-P. Kong for the 830 A Fab. A.H. is the recipient of an amfAR Mathilde Krim Fellowship in Basic Biomedical Research (108501-53-RKNT) and was also supported by a Phase II amfAR research grant 109285-58-RKVA for independent investigators. Support for this work was also provided by grants from the NIH to J.G.S. (grants AI24755, AI124982, P01 GM56550, and AI100645), W.M. (grants GM116654 and P01 GM56550), A.B.S. (P01 GM 56550) and S.C.B. (grant GM098859).

## Author contributions

A.H. and C.G. performed the mutagenesis, antibody binding, and virus inhibition experiments; A.S. and E.F. performed the thermodynamics studies; M.F. and A.B.S. synthesized the chemical probes; P.D.K. and M.P. solved the structure of BG505SO-SIP.664 with BMS-626529. F.M. and C.A. performed the molecular modeling; X.M. and W.M. performed the smFRET experiments; and A.H. and J.G.S. analyzed the data and wrote the paper.

## Additional information

**Competing interests:** The authors declare no competing financial interests.

**Change history:** A correction to this article has been published and is linked from the HTML version of this paper.

