## [Peer Review file · Nature Communications]

Editorial Note: *This manuscript has been previously reviewed at another journal that is not operating a transparent peer review scheme. This document only contains reviewer comments and rebuttal letters for versions considered at Nature Communications. Mentions of prior referee reports have been redacted.*

Reviewer #1 (Remarks to the Author):

The revised paper is improved compared with the original, [redacted]. I still believe that this paper is more appropriate for a journal that specializes in HIV research than a general journal such as Nature Communications since it would be hard for those without a background in HIV-1 Env structures to understand the significance of the findings. [Redacted]. However, I will leave the decision of whether to accept this paper in Nature Communications to the discretion of the editors and other reviewers. In any case, I believe there are remaining issues that should be addressed before publication. Referring to the authors' response to my original review, I have the following comments (in each case, preceded by the words "Reviewer response.")

[Redacted]

Reviewer #2 (Remarks to the Author):

The revised manuscript is significantly improved and is acceptable to this reviewer. The authors should be commended on their comprehensive revisions, which increased the clarity and impact of the manuscript. There are just two minor issues with Fig. 1 that should be addressed.

1. Fig. 1A: CAP120.2.00.E8, ZM109 (was this Env clone F.PB4?), ZM53M.PB12, and Du422.1 envelopes were derived from acute/early HIV-1 infections, but before the more formal description of 'transmitted/founder' variants. I would leave it up to the authors as to whether they want to indicate this information.

2. Fig. 1B: KB9 should be clade B and CAP210 should be clade C. Also, the period is in the wrong place for CAP210 (see panel A vs. B).

Reviewers' comments:

Reviewer #1 (Remarks to the Author):

The revised paper is improved compared with the original, *[redacted]*. I still believe that this paper is more appropriate for a journal that specializes in HIV research than a general journal such as Nature Communications since it would be hard for those without a background in HIV-1 Env structures to understand the significance of the findings. *[Redacted]*. However, I will leave the decision of whether to accept this paper in Nature Communications to the discretion of the editors and other reviewers. In any case, I believe there are remaining issues that should be addressed before publication. Referring to the authors' response to my original review, I have the following comments (in each case, preceded by the words "Reviewer response.")

Authors' response 2: As we responded before, although some background knowledge of the HIV-1 envelope glycoproteins will help the reader to appreciate our work fully, there are general principles presented in the manuscript related to the function of viral fusion proteins, the sensitivity of viral envelope glycoproteins to broadly neutralizing antibodies, and the use of chemical probes to elucidate biological mechanisms of action that should be of interest to a more general reader. In addition, our manuscript is consistent with the aim of Nature Communications to publish papers that "represent important advances of significance to specialists within each field".

[Redacted]

Reviewer #2 (Remarks to the Author):

The revised manuscript is significantly improved and is acceptable to this reviewer. The authors should be commended on their comprehensive revisions, which increased the clarity and impact of the manuscript. There are just two minor issues with Fig. 1 that should be addressed.

1. Fig. 1A: CAP120.2.00.E8, ZM109 (was this Env clone F.PB4?), ZM53M.PB12, and Du422.1 envelopes were derived from acute/early HIV-1 infections, but before the more formal description of 'transmitted/founder' variants. I would leave it up to the authors as to whether they want to indicate this information.

2. Fig. 1B: KB9 should be clade B and CAP210 should be clade C. Also, the period is in the wrong place for CAP210 (see panel A vs. B).

Authors' Response 2: We were pleased to see that the reviewer felt that the revisions increased the clarity and impact of the revised manuscript. As suggested, we added the acute/early designation to the legend of Fig. 1. We also corrected the annotation of KB9 and CAP210 as clade B and C in Fig. 1b.